# Body Composition Interactions with Physical Fitness: A Cross-Sectional Study in Youth Soccer Players

**DOI:** 10.3390/ijerph19063598

**Published:** 2022-03-18

**Authors:** César Leão, Ana Filipa Silva, Georgian Badicu, Filipe Manuel Clemente, Roberto Carvutto, Gianpiero Greco, Stefania Cataldi, Francesco Fischetti

**Affiliations:** 1Escola Superior Desporto e Lazer, Instituto Politécnico de Viana do Castelo, Rua Escola Industrial e Comercial de Nun’Álvares, 4900-347 Viana do Castelo, Portugal; cleao@esdl.ipvc.pt (C.L.); anafilsilva@gmail.com (A.F.S.); filipe.clemente5@gmail.com (F.M.C.); 2Research Center in Sports Performance, Recreation, Innovation and Technology (SPRINT), 4960-320 Melgaço, Portugal; 3The Research Centre in Sports Sciences, Health Sciences and Human Development (CIDESD), 5001-801 Vila Real, Portugal; 4Department of Physical Education and Special Motricity, Faculty of Physical Education and Mountain Sports, Transilvania University of Braşov, 500068 Braşov, Romania; georgian.badicu@unitbv.ro; 5Instituto de Telecomunicações, Delegação da Covilhã, 1049-001 Lisboa, Portugal; 6Department of Basic Medical Sciences, Neuroscience and Sense Organs, University of Study of Bari, 70124 Bari, Italy; roberto.carvutto@uniba.it (R.C.); gianpierogreco.phd@yahoo.com (G.G.); francesco.fischetti@uniba.it (F.F.)

**Keywords:** football, body fat distribution, athletic performance, physical fitness

## Abstract

This study aimed to: (i) analyze fat mass and physical fitness variations among age-groups and playing positions, and (ii) explore the relationship between fat mass and physical fitness in youth male soccer players. A total of 66 players from under-16, under-17, and under-19 were tested. Body mass, skinfolds, countermovement jump (CMJ), single-leg triple hop jump (SLTH), bilateral triple hop jump (BTH), and yo-yo intermittent recovery Level 2 (YYIR-2) were assessed. A two- and one-way ANOVA were conducted, and the effect size was measured. Interactions were found in skin folds and fat mass. The under-19 group was taller, heavier, with a greater BMI and muscle mass than the under-16 group. They also exceeded the under-16 and under-17 in SLTH, BTH, and YYIRT-2. The under-17 group jumped higher and longer than under-16 group. Goalkeepers were taller and heavier than the midfielders. Central defenders were taller and had more muscle mass than midfielders and were heavier than the midfielders and wingers. The wingers jumped higher than the midfielders and showed better YYIRT-2. BMI was small correlated with YYIRT-2 and moderately with CMJ. Fat mass had a moderate negative correlation with CMJ and YYIRT-2. Muscle mass largely correlated with CMJ, UTH, very large with BTH and moderate with YYIRT-2. Summarily, with increasing age, better performances and body compositions were registered. Muscle mass better influences performance than body fat. Body composition can distinguish players positions.

## 1. Introduction

To achieve success in soccer, several features including physical characteristics, physiological capacities, as well as motivation and sports-specific knowledge are needed [1,2,3]. Considering that soccer players are normally selected to be included in academies at very young ages, knowing those features and their influencing factors seems to play an important role. Some studies have been conducted that characterize elite players and describe the factors that influence the progression of young soccer players [2,4,5,6]. Nevertheless, in some previous studies, the playing positions were not considered, leading to a potential confounding variable [7]. Indeed, team sports are more complex to analyze than individual sports, where discrete objective measures of performance are more easily observed [8].

During a decade, the anthropometric features of top-level soccer players were analyzed, and it was registered that there was increasing height and body mass among adult professional players (approximately 2 cm and 1.5 kg, respectively) [9]. This could mean that anthropometric characteristics influence performance in this sport. This assumption has already been highlighted, i.e., it was speculated that optimal body composition and greater anthropometric variables could be an advantage since it could help to develop levels of muscle force and power, which leads to a more efficient movement [10,11]. In fact, in the study by Bongiovanni et al. [12], it was noticed that anthropometric features are important predictors of sprint performance and aerobic fitness in a sample of youth elite soccer players. Also, in the Esco et al. [13] study, it was observed that players with a lower fat mass better performed the maximal incremental running test and vertical jump-and-reach task. Those authors added that elevated level of fat tissue together with a lower level of muscle mass may negatively affect physical performance in youth soccer [13].

The literature that focused on young soccer players revealed that child and adolescent players exhibited a significantly lower body fat than the reference population [13,14,15]. Also, body composition registers visible changes during the growth spurt, which occurs in boys around 14 years of age [16]. In addition, Nikolaidis et al. [17] noticed that the age corresponding to U17, seems to determine a turning point in adolescence, in which significant changes in fat mass and fat-free mass were observed [17]. In fact, associations between age and body composition across adolescence were already described, however, no consensus regarding the direction of this association were found, since both increase and decrease of fat mass across adolescence has been reported [17,18,19].

Considering that soccer is characterized as an intermittent activity, with events changing every 3–5 s, it is well understood that players should be prepared for intense actions involving jumps, turns, tackles, high-speed runs, and sprints [20,21,22]. Studies have observed that aerobic fitness, agility, and explosive power in the lower body are important for achieving higher performances [23]. However, considering the changes that occur with growth, and especially with the maturation process, previous research has suggested that different physical performance characteristics become apparent in different age-groups [24,25]. For instance, in the early ages sprinting ability seems to be more crucial (ages between 10 to 14 years; [6,7,8,9,10,11,12,13,14,15,16,17,18,19,20,21,22]), whereas aerobic endurance was found to be more important in older players (15- and 16-year; [6]). These different levels of importance can also be influenced by the development of the players’ body composition.

Considering the above-mentioned knowledge, the purpose of the present study was two-fold: (i) to analyze the variations of fat mass and physical fitness between age-groups and playing positions, and (ii) to investigate the relationship between fat mass and physical fitness in youth male soccer players.

## 2. Materials and Methods

This study followed an observational design. A total of 87 youth soccer players (age: 16.5 ± 1.1 years; height: 174 ± 0.1 cm; body mass: 66.9 ± 8.3 kg) from under-16 (*n* = 29), under-17 (*n* = 28), and under-19 (*n* = 30) age-groups were included in this study. They were grouped by age-group and by position. The inclusion criteria consisted of (1) all the physical assessments that were performed at the beginning of the season, (2) the absence of injury during the time of the assessments, and the absence of any injury in the previous month before the assessments. After applying the criteria, 66 youth soccer players (age: 16.6 ± 1.1 years; height: 175 ± 0.1 cm; body mass: 66.8 ± 7.9 kg) from under-16 (*n* = 21), under-17 (*n* = 19), and under-19 (*n* = 26) age-groups were included in our analysis. Figure 1 describes the anthropometric characteristics. The study was conducted in accordance with the Declaration of Helsinki and approved by the Escola Superior de Desporto e Lazer ethical committee with the code CTC-ESDL-CE001-2021.

All the tests were conducted in the first week of training for every age-group in the season 2021–2022 (U19 and U17: 19 to 23 of July; U16: 9–15 August).

### 2.1. Anthropometry

All the tests were overseen in an appropriate room, just before the participants’ training. All the athletes wore light clothing and stood barefoot. They had their body mass assessed to the nearest 0.1 kg with a digital scale (Prozis SmartScale, Prozis, Madeira, Portugal) and their height measured to the nearest 0.1 cm with a portable stadiometer (Seca 217, Hamburg, Germany). Following the guidelines of the International Society for the Advancement of Kinanthopometry [26], eight skinfolds (triceps, subscapular, biceps, suprailiac, abdominal, supraspinal, thigh, and calf) were measured twice (at 0.1 mm) with a Harpenden caliper (British Indicators, Ltd., London, UK). The mean value of the measurements was considered and the sum of the eight skinfolds was calculated. The equation of Slaughter [27] was applied to estimate body fat. Additionally, the equation of Poortmans [28] was used to estimate the muscle mass. All the measurements were performed by a Level 2 ISAK certified tester. It should be noted that all the athletes were fully mature and involved in the official competition, hence the utilization of these equations.

### 2.2. Vertical Jumps

After a standardized warm-up period of 5 to 10 min, the athletes carry on 3 repetitions of the countermovement jump (CMJ). They assumed an upright position with the arms locked on the waist throughout the entire motion of the jump. After hearing a signal from the coach, they flexed the knees and jumped, in a single movement, as high as possible. The legs must stay straight, observing plantar flexion. The subjects were instructed to land in the same starting point and to keep the legs straight upon landing, in order to avoid knee bending and the alteration of the measurements. Each subject was given at least 60 s for rest in-between jumps. All the trials were performed by athletes that were positioned on a contact platform (Chronojump) that was attached to hardware (Chronopic^®^, Chronojump Boscosystem, Barcelona, Spain). The hardware was connected to a computer that displayed the vertical jump values (cm) from a free software (Chronojump Boscosystem Software, Barcelona, Spain). The best of the 3 trials was recorded to the nearest 0.1 cm. All the jumps were performed right before training with the participants wearing sports shoes.

### 2.3. Horizontal Jumps

An open area was set, and an 8-m measuring tape was put on the floor. A strip was placed perpendicular, creating the starting line. A standardized warm-up was executed by the athletes, after which they proceeded to do 3 trials of each jump, single-leg triple hop jump (SLTH) and bilateral triple hop jump (BTH), with 1 min, at minimum, of rest between the attempts.

To execute the SLTH the athlete would start standing on the designated leg, touching the starting line with his feet. He would then perform 3 consecutive maximal hops. They were instructed to hold the landing foot on the last jump to the place that they landed, although they could use upper extremity movement to keep balance. The distance from the starting line to the point where the subject’s heel landed was registered to the nearest 0.1 cm. After the conclusion of all the trials, the mean result of the 3 jumps for each leg was calculated.

The BTH started with the athlete touching the starting line with one toe that was chosen by the participant. He would then, without balance run, perform a triple jump. They were instructed to hold the landing foot on the last jump to the place that they landed, although they could use upper extremity movement to keep balance. The distance from the starting line to the point where the subject’s heel landed was registered to the nearest 0.1 cm. After the conclusion of all the trials, the mean result of the 3 jumps was calculated.

All the jumps, SLTH and BTH were performed right before training, with the participants wearing sports shoes.

### 2.4. Yo-Yo Intermittent Recovery Level 2 (YYIR2)

Following a warm-up period, the participants perform repeated 2 × 20-m runs at progressively increasing speed, intermitted by 10-s periods of active recovery (2 × 5 m) [29]. The test was performed until total exhaustion of the participant was reached (i.e., as maximal performance test). The YYIR2 test started at a higher speed level and two initial runs of 13 and 15 km·h^−1^, respectively, followed by two runs at 16 km·h^−1^, three runs at 16.5 km·h^−1^, 4 runs at 17.0 km·h^−1^, proceeding with stepwise 0.5 km·h^−1^ speed increments after every 8 running bouts until exhaustion. The pace was controlled by an automated acoustic device, indicating start, turn, and stop. The test was finalized when the athlete failed to reach the finishing line in time two times or if, due to perceived exhaustion, the test was discontinued.

### 2.5. Statistical Procedures

Descriptive statistics are presented in the results in the form of the mean and the standard deviation. Exploratory inspection of the -outliers did not reveal significant variations. The Kolmogorov–Smirnov and Levene’s test revealed normality (*p* > 0.05) and homogeneity (*p* > 0.05) of the different anthropometry, body composition, and physical fitness outcomes. A two-way ANOVA was used to test the possible interactions between age-group and playing positions. The effect size of the two-way ANOVA was executed using the partial eta squared. A one-way ANOVA was conducted to analyze the variations of outcomes between the age-groups and playing positions. The effect size of the one-way ANOVA was executed using eta squared. The Tukey HSD test was used as post hoc test after the one-way ANOVA, revealing significant differences between the groups. The correlations between body composition and physical fitness outcomes were examined using Pearson’s product-moment correlation test. The magnitude of the correlations was settled based on the following thresholds [30]: 0.0–0.1, trivial; 0.1–0.3, small; 0.3–0.5, moderate; 0.5–0.7, large; 0.7–0.9, very large; and 0.9–1.0, nearly perfect. All the statistical tests were executed using the Statistical Package for the Social Sciences (SPSS, version 28.0.0.0., IBM, Boston, IL, USA) for a *p* < 0.05.

## 3. Results

Descriptive statistics of anthropometry and body composition outcomes can be found in Figure 2. The two-way ANOVA revealed significant interactions (age-group * playing position) on the sum of skinfolds (*p* = 0.005; $\eta_{p}^{2}$ = 0.366). No significant interactions (age-group * playing position) were found in fat mass (*p* = 0.064; $\eta_{p}^{2}$ = 0.270), body mass (*p* = 0.764; $\eta_{p}^{2}$ = 0.106), height (*p* = 0.354; $\eta_{p}^{2}$ = 0.176), body mass index (*p* = 0.308; $\eta_{p}^{2}$ = 0.186), and muscle mass (*p* = 0.618; $\eta_{p}^{2}$ = 0.130).

Descriptive statistics of physical fitness outcomes can be found in Figure 3. No significant interactions (age-groups * playing positions) were found in CMJ (*p* = 0.543; $\eta_{p}^{2}$ = 0.143), TH bilateral (*p* = 0.850; $\eta_{p}^{2}$ = 0.089), TH right leg (*p* = 0.328; $\eta_{p}^{2}$ = 0.181), TH left leg (*p* = 0.977; $\eta_{p}^{2}$ = 0.050), and YYIRT (*p* = 0.270; $\eta_{p}^{2}$ = 0.194).

Descriptive statistics of anthropometry, body composition, and physical fitness outcomes organized per age-group can be found in Table 1. The under-19’s were significantly taller (+0.1 cm; *p* < 0.001) and heavier (+10.0 kg; *p* < 0.001) than under-16’s. Significantly greater BMI (+1.3 kg/m^2^; *p* = 0.041) and muscle mass (+5.2 kg; *p* < 0.001) were also found in the under-19’s in comparison to the under-16’s. Regarding the physical fitness outcomes, it was found that the under-19’s jumped significantly higher in CMJ than the under-16’s (+9.4 cm; *p* < 0.001) and the under-17’s (+5.3 cm; *p* < 0.001). Moreover, CMJ was also significantly higher in the under-17’s than in the under-16’s (+4.1 cm; *p* = 0.006). Considering the triple hop bilateral, the under-19 group jumped significantly longer than the under-16 group (+1.1 m; *p* < 0.001) and under-17 group (+0.5 m; *p* = 0.003), while the under-17’s jumped significantly longer than the under-16’s (+0.6 m; *p* < 0.001). Regarding the unilateral triple hop, it was found that the under-19’s jumped significantly longer than the under-16’s in both the right (+0.7 m; *p* < 0.001) and left (+0.7 m; *p* < 0.001) legs. Finally, the under-19 group showed a significantly greater YYIRT performance than the under-16’s (+224.1 m; *p* < 0.001) and the under-17’s (+99.7 m; *p* = 0.025), while the under-17 group presented significantly greater YYIRT than the under-16’s (+124.4; *p* = 0.006).

Table 2 presents the descriptive statistics of anthropometry, body composition, and physical fitness outcomes organized by playing position. The goalkeepers were significantly taller than the midfielders (+1 cm; *p* = 0.011) and wingers (+1 cm; *p* = 0.024), while the central defenders were significantly taller than the midfielders (+1 cm; *p* = 0.023). The goalkeepers were significantly heavier than the midfielders (+11.4 kg; *p* = 0.044), while the central defenders were significantly heavier than the midfielders (+9.8 kg; *p* = 0.003) and wingers (+9 kg; *p* = 0.035). The wingers jumped significantly higher than the midfielders (+5.6 cm; *p* = 0.051). The wingers had significantly better performance at YYIRT than the forwards (+227.3 m; *p* = 0.033).

The correlation levels between body composition and physical fitness outcomes for the overall players can be observed in Table 3. The body mass index had small magnitude correlations with YYIRT (r = 0.261; *p* = 0.036) and moderate correlations with CMJ (r = 0.360; *p* = 0.003). The fat mass had moderate and significant negative correlations with CMJ (r = −0.315; *p* = 0.011), bilateral TH (r = −0.323; *p* = 0.009), and left TH (r = −0.260; *p* = 0.036). The muscle mass had large correlations with bilateral TH (r = 0.547) and moderate correlations with CMJ (r = 0.498; *p* < 0.001), unilateral TH in right (r = 0.440; *p* < 0.001), and left (r = 0.439; *p* < 0.001) legs, and YYIRT (r = 0.498; *p* < 0.001).

## 4. Discussion

The current study aimed to explore the variations of body fat mass, muscle mass, and physical fitness between age-groups and playing positions and to investigate the relationship between body composition and physical fitness in youth male soccer players. The main results of our study showed that with increasing age we see a reduction in the percentage of body fat and an increase in the muscle mass, which indicates an improvement in body composition. Moreover, it was found that muscle mass had a greater magnitude of correlations with physical fitness outcomes than body fat mass.

With respect to anthropometric values, our study revealed an increase in the stature and body mass with advancing age, while in the opposite direction we see a decrease in the percentage body fat. In this sense, we see that the characteristics of the participants are within what is found in other studies [4,5,6,17,31,32,33,34] describing the same kind of results. Although the values follow the same pattern in relation to the absolute values, we notice some discrepancy. In some studies we have weight values that are higher than those of our participants in all age groups [4,35,36], while in other studies these values are comparable to ours [8,34] or are even lower [37]. The same is true for height, although we found a greater number of studies reporting a height that was greater than that of our sample [4,33,35,36] compared to an equal [8] or smaller height [37]. With regard to fat mass, we have a greater discrepancy of results, and although it is possible to find samples with similar values [8], in general there is a great variability of values, being possible to find higher values [4,34,36], but also lower values [35,37] than those that were found by our group of researchers.

This discrepancy in values that is found in the literature can be explained by three reasons. First, the genetic profile of the population of origin of the participating athletes, which may influence the physical characteristics of different samples. Second, the different level of play that players are involved in [38]. And finally, the use of different assessment methods or different equations to estimate the percentage of body fat mass could lead to the different values that were found [39].

Considering the physiological specificities of football, it is possible to find differences in anthropometric characteristics in the various positions also [38,40]. When analyzing our results, we concluded that goalkeepers are taller, heavier, and with a higher percentage of body fat and muscle mass compared to the other positions. These results are consistent with what we found in the literature, not only when we think about the group age of our participants [8,31,41,42,43,44,45,46], but also for adult players [47,48,49]. Additionally, and also in line with is found in the literature, we found that the central defenders are the ones with greater stature and greater body mass, despite some homogenization between the field positions. This is mostly in line with what we would expect, due to the general playing demands of the respective positions in the field.

It is well known that body composition, namely the percentage of body fat, has a negative impact on aerobic and anaerobic capacity, strength, power, and speed [2,50,51,52,53], as well as the ability to perform high-intensity and maximum-speed running [54]. Additionally, the effect that physical conditioning has in the ability to jump vertically and in the ability to perform specific tasks quickly is also known [55,56,57,58]. In the same vein, there is some evidence that muscle mass plays a crucial role in improving physical performance [59], even in young soccer players [60].

In that regard, and with the corroboration with what is found in other studies [61,62,63,64], we noticed a negative relationship between the body fat percentage and the results in the tests that were performed by the players. Moreover, and as a more novel finding, we could establish a stronger positive relationship between the results of the tests and total muscle mass.

It is important to note that the decrease in the percentage of body fat that was observed at this age is more a consequence of the gain in muscle mass that is attained by the athletes than the loss of fat mass. We can speculate that the increase in muscle mass could have led to a change in the percentage of body fat mass without decreasing the absolute value of fat mass, something that was already concluded by other researcher [42]. This fact also is in agreement with what was observed by Hannon et al. [36], and, in a way, corroborates the idea that as the player goes through their development process, muscle mass can assume a greater importance relative to body fat regarding their performance in soccer players.

This study had some limitations. To begin with, this study only incorporates a small number of participants. Also, the participants were selected by convenience sampling; thus, findings may be the result of a contextual factor. However, the results that are presented can be supported by previous works, which provide some confidence regarding the generalization. Moreover, no gold-standard method to analyze the body composition was used (e.g., dual-energy X-ray absorptiometry), however skinfolds have good levels of concurrent validity with gold-standard methods which provide confidence about the accuracy of the data. Future research should consider analyzing how muscle mass can enhance physical fitness outcomes and the interactions with the type of training that the players are exposed to. Even so, this study opens the possibilities of new lines of investigation that focus on the variables to be taken into account in the anthropometric monitoring of young soccer athletes in relation to performance.

As a practical implication, this study showed that between the ages of 15 and 18, results in physical tests, as well as body composition, improve in soccer players, with a strong relationship with the increase in muscle mass. Considering this, we can think that muscle mass can be more important to enhance physical fitness in young soccer players in the final stages of their development. Despite that, it is important to note that muscle mass that is gained over the ages must be followed by a correct neural-based training for improving the maximization of contractibility of muscle mass to produce powerful actions.

## 5. Conclusions

The present study showed that with the increasing age, better physical performances and body compositions were observed in soccer players. Both the muscle mass (positive) and the body fat percentage (negative) influenced that performance, despite the first showing greater influence. The comparison between the players’ position showed that the goalkeepers were taller and heavier than the midfielders, while the central defenders were taller than the midfielders and heavier than midfielders and wingers. The central defenders had more lean body mass than the midfielders. In addition, the wingers jumped higher than the midfielders, and had a better performance at YYIRT than the forwards.

In the future, research should focus on trying to understand the impact of different physical conditioning programs that are applied to young football players on the development of muscle mass and the impact that this will have on their performance in physical tests, and on their chances of having a successful football career.

## Figures and Tables

**Figure 1 ijerph-19-03598-f001:**
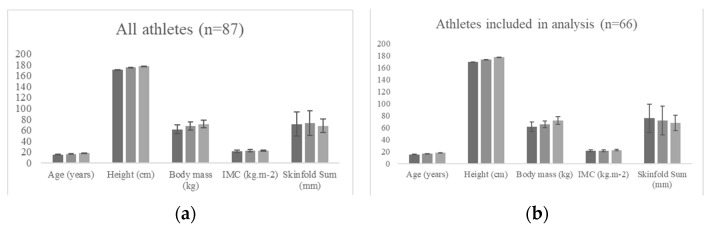
Characteristics of the observed population. (**a**) All athletes: U-16 (*n* = 29), U-17 (*n* = 28), and U-19 (*n* = 30). (**b**) Athletes included in the analysis: U-16 (*n* = 21), U-17 (*n* = 19), and U-19 (*n* = 26).

**Figure 2 ijerph-19-03598-f002:**
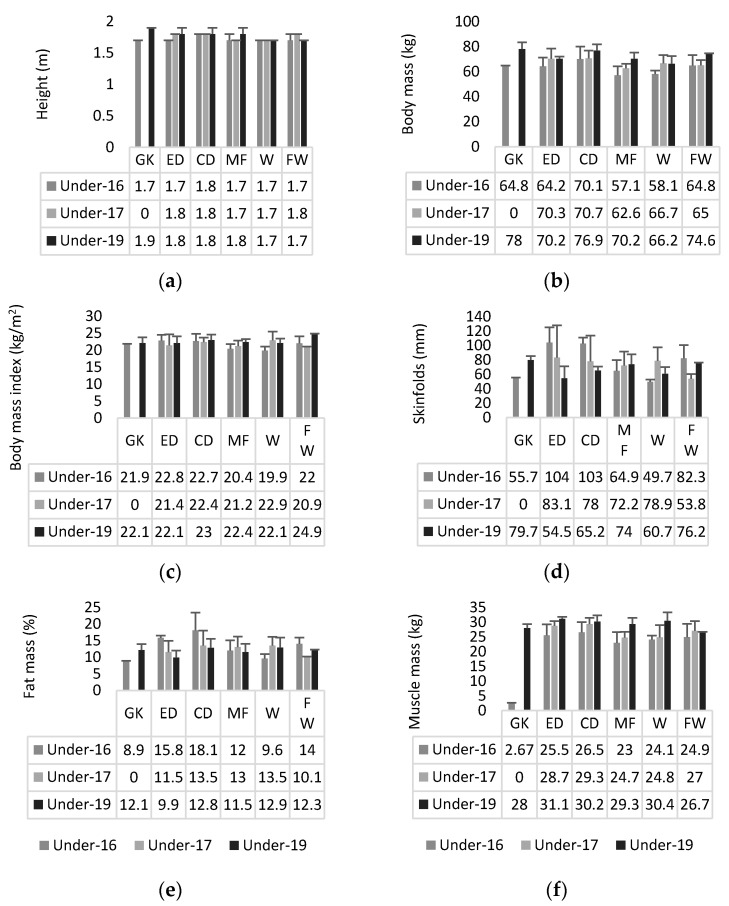
Descriptive statistics (mean ± standard deviation) of (**a**) height, (**b**) body mass, (**c**) body mass index, (**d**) sum of skinfolds, (**e**) fat mass, and (**f**) muscle mass for each age-group and playing position. GK—goalkeeper; ED—external defender; CD—central defender; MF—midfielder; W—winger; FW—Forward.

**Figure 3 ijerph-19-03598-f003:**
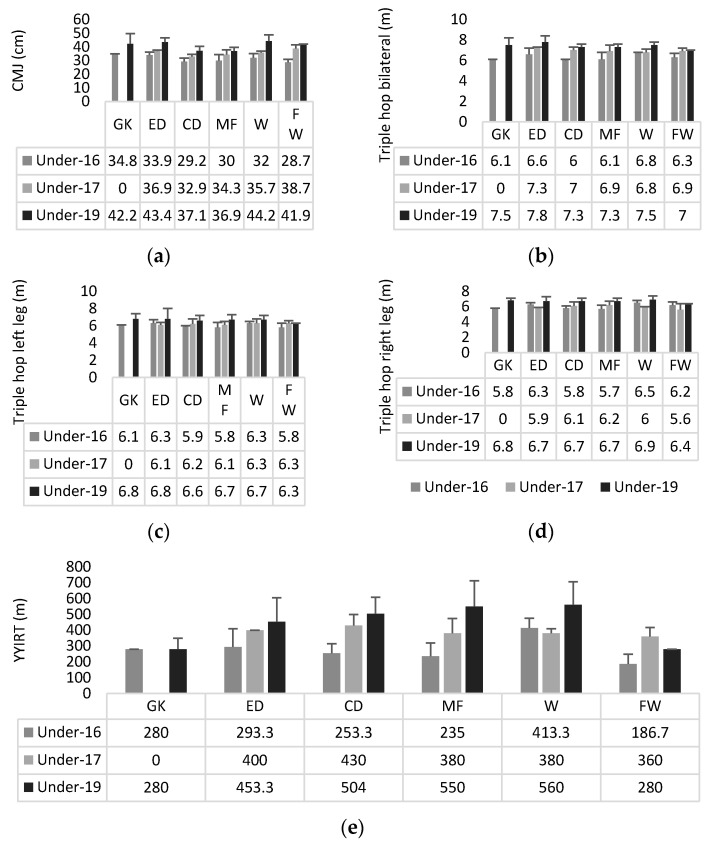
Descriptive statistics (mean ± standard deviation) of (**a**) CMJ, (**b**) triple hop bilateral, (**c**) triple hop left, (**d**) triple hop right, (**e**) yo-yo intermittent recovery test—level 2. GK—goalkeeper; ED—external defender; CD—central defender; MF—midfielder; W—winger; FW—Forward.

**Table 1 ijerph-19-03598-t001:** Descriptive statistics (mean and standard deviation) of anthropometry, body composition, and physical fitness outcomes organized per age-group.

	Under-16 (*n* = 21)	Under-17 (*n* = 18)	Under-19 (*n* = 26)	*p*	Effect Size $\eta^{2}$
Height (cm)	169.8 ± 6.1	174.6 ± 4.6	178.5 ± 7.0	<0.001 **	0.271
BM (kg)	61.6 ± 8.0	66.0 ± 5.7	71.6 ± 6.3	<0.001 **	0.297
BMI (kg/m^2^)	21.3 ± 1.9	21.7 ± 1.7	22.5 ± 1.4	0.044 *	0.096
Skinfolds (mm)	75.8 ± 23.8	73.4 ± 24.2	67.7 ± 13.0	0.380	0.031
Fat mass (%)	13.2 ± 3.9	12.6 ± 3.2	12.0 ± 2.5	0.431	0.027
Muscle mass (kg)	24.5 ± 3.4	26.4 ± 2.9	29.7 ± 2.2	<0.001 **	0.401
CMJ (cm)	30.8 ± 3.6	34.9 ± 3.1	40.2 ± 4.9	<0.001 **	0.507
TH bilateral (m)	6.3 ± 0.5	6.9 ± 0.4	7.4 ± 0.4	<0.001 **	0.540
TH right leg (m)	6.0 ± 0.5	6.1 ± 0.5	6.7 ± 0.4	<0.001 **	0.395
TH left leg (m)	6.0 ± 0.5	6.2 ± 0.4	6.7 ± 0.6	<0.001 **	0.272
YYIRT (m)	266.7 ± 98.3	391.1 ± 72.0	490.8 ± 158.6	<0.001 **	0.391

BM: body mass; BMI: body mass index; CMJ: countermovement jump; TH: triple hop; YYIRT: Yo-Yo intermittent recovery test—level 2; * significant at *p* < 0.05; ** significant at *p* < 0.01.

**Table 2 ijerph-19-03598-t002:** Descriptive statistics (mean and standard deviation) of anthropometry, body composition, and physical fitness outcomes organized by playing position.

	GK (*n* = 4)	ED (*n* = 8)	CD (*n* = 12)	MF (*n* = 24)	WG (*n* = 11)	FW (*n* = 6)	*p*	Effect Size $\eta^{2}$
Height (cm)	184.0 ± 8.3	175.1 ± 7.3	179.3 ± 6.0	172.0 ± 6.8	172.1 ± 3.9	173.3 ± 5.6	0.002 **	0.266
BM (kg)	74.7 ± 7.9	68.0 ± 5.9	73.1 ± 6.9	63.3 ± 7.5	64.1 ± 6.3	66.5 ± 7.0	0.001 **	0.277
BMI (kg/m^2^)	22.0 ± 1.4	22.2 ± 2.0	22.8 ± 1.5	21.3 ± 1.5	21.6 ± 1.8	22.1 ± 2.0	0.277	0.099
Skinfolds (mm)	73.7 ± 12.9	80.3 ± 32.0	78.8 ± 24.7	70.4 ± 16.0	61.0 ± 13.5	71.8 ± 18.5	0.303	0.095
Fat mass (%)	11.3 ± 2.2	12.5 ± 3.3	14.4 ± 4.3	12.2 ± 2.9	12.1 ± 2.8	12.4 ± 2.3	0.413	0.080
Muscle mass (kg)	27.7 ± 1.2	28.4 ± 3.3	29.0 ± 2.7	25.7 ± 3.7	27.7 ± 4.0	25.9 ± 3.4	0.095	0.144
CMJ (cm)	40.4 ± 7.1	38.2 ± 4.9	33.7 ± 4.1	33.7 ± 4.5	39.3 ± 6.7	34.3 ± 6.4	0.014 *	0.211
TH bilateral (m)	7.1 ± 0.9	7.2 ± 0.7	6.9 ± 0.6	6.8 ± 0.7	7.2 ± 0.4	6.6 ± 0.5	0.304	0.095
TH right leg (m)	6.6 ± 0.5	6.3 ± 0.4	6.3 ± 0.5	6.2 ± 0.6	6.6 ± 0.5	6.0 ± 0.5	0.250	0.104
TH left leg (m)	6.6 ± 0.6	6.4 ± 0.7	6.3 ± 0.6	6.2 ± 0.6	6.5 ± 0.5	6.1 ± 0.5	0.537	0.065
YYIRT (m)	280.0 ± 56.6	380.0 ± 126.5	416.7 ± 129.3	388.3 ± 173.5	487.3 ± 136.0	260.0 ± 97.2	0.039 *	0.176

GK: goalkeeper; ED: external defender; CD: central defender; MF: midfielder; WG: winger; FW: central forward/striker; BM: body mass; BMI: body mass index; CMJ: countermovement jump; TH: triple hop; YYIRT: Yo-Yo intermittent recovery test—level 2; * significant at *p* < 0.05; ** significant at *p* < 0.01.

**Table 3 ijerph-19-03598-t003:** Pearson correlation coefficient (correlation coefficient; (95%confidence interval, minimum; maximum)) between body composition and physical fitness outcomes.

	CMJ (cm)	TH Bilateral (m)	TH Right Leg (m)	TH Left Leg (m)	YYIRT (m)
BMI (kg/m^2^)	0.360 ** (0.128; 0.555)	0.194 (−0.053; 0.418)	0.202 (−0.044; 0.425)	0.190 (−0.056; 0.415)	0.261 * (0.018; 0.475)
Fat mass (%)	−0.315 * (−0.519; −0.077)	–0.323 ** (−0.526; −0.086)	–0.151 (−0.381; 0.096)	−0.260 * (−0.474; −0.018)	−0.240 (−0.457; 0.004)
Muscle mass (kg)	0.573 * (0.382; 0.717)	0.547 ** (0.350; 0.698)	0.440 ** (0.220; 0.618)	0.439 ** (0.219; 0.617)	0.498 ** (0.289; 0.661)

BMI: body mass index; CMJ: countermovement jump; TH: triple hop; YYIRT: Yo-Yo intermittent recovery test—level 2; * significant at *p* < 0.05; ** significant at *p* < 0.01.

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
