# Peer review of "Body Composition Interactions with Physical Fitness: A Cross-Sectional Study in Youth Soccer Players"

_ijerph, 2022, doi:10.3390/ijerph19063598_

Round 1

Reviewer 1 Report

The abstract is well structured and informative.

The method is precise and clearly written.

All measurements and tests can be repeated.

Statistical procedures are appropriate.

The results are accurately presented and adequately interpreted.

The tables are informative and clear.

The discussion is the weakest link in this study.

Line 261-283 has too many stories of other studies.

There should be a comparison, but detailing other people's studies is superfluous.

Lines 284-296 also have a lot of unnecessary.

Line 299-303 does not need to be detailed about the tests. They should not be written about in the discussion at all.

They neglected the story of the connection between body composition and fitness. They do not give any explanation why.

The discussion is very weak and boring to read. There is a lot of work to be done.

Some references are not in the standard style (11, 14, 24, 65).

Reviewer 2 Report

Thank you for the opportunity to review the interesting topic “Body Composition Interactions with Physical Fitness: A Cross Sectional Study in Youth Soccer Players”.

The following observations can help to improve the manuscript:

1. It is recommended to transfer the results of the study (lines 87-95) to the Results. Figure 1 should be moved to Results. It should be noted that the data shown in Figure 1 overlap with the results of Table 1 (age, height, body mass, skinfold sum).

2. The information in lines 98-100 must be moved to lines 354-355. The information in line 352 should be specified by the protocol number and the exact date of approval in accordance with the recommendations of IJERPH.

3. It seems necessary to specify the numbers of caliper calibration certificate and ISO (national standards for achieving calibration). It is necessary to revise the procedure for measuring the skinfolds (whether the measurements were carried out in a certified laboratory, who carried out the measurements, how many samples were tested, etc.). Similarly, it is necessary to clarify the information in subsections 2.2, 2.3, 2.4 of the paper.

4. Line 151: citation for literature must be unified: „(Bangsbo et al., 2008)“ must be changed to [51]. There seems to be a need for re-numbering references according to the citation order.

5. Lines 153-154: There is an obscure marking (namely „15 km·h−1“, „16 km·h−1“, „16.5 km·h−1“, „17.0 km·h−1“, „0.5 km·h−1“) that needs to be corrected.

6. Line 186: Figure 2 is non-informative and needs to be converted into a table so that the reader can see the exact data. Similar procedures are recommended for Figure 3.

7. Line 247: It is recommended to clarify the title of Table 3: “average; [95%Confidence interval, Minimum; Maximum)”. I would suggest that “average” be changed to “the correlation coefficient” You shouldn't write „Minimum“, „Maximum“. The same symbols such as “[„ and „]“ (not „)“) must be used in the title of Table 3.

8. Lines 256-258: It is necessary to clarify the sentence written by the author.

9. Line 320-327: It seems necessary to insert a limitation related to a small sample size of the study. I would recommend changing the title and abandoning the expression „A Cross Sectional Study“. Due to the small sample size of the study, it might be optimal to write as „A Pilot Study“. Nevertheless, I am interested in the paper.

10. The authors indicated in the Discussion that their results coincide with those previously published by other researchers. There seems to be a lack of response to the question of why the study is new and relevant. Practical recommendations could be more targeted and precise. Practical proposals could be combined with Conclusions. I would also suggest writing „lean body mass“ instead of „the lean mass“, etc.

11. I would recommend that authors re-examine the references and edit it as recommended by IJERPH.

Kind Regards

Reviewer 3 Report

Thank you for your submitted manuscript entitled, “Body Composition Interactions with Physical Fitness: A Cross-2 Sectional Study in Youth Soccer Players 3’’. Overall, this is a well-written manuscript.

- Experimental question and novelty of the study: What is the question being answered? What did the authors want to show?Since there have been many studies that describe the aims of the study, why did the authors conduct this study?

-What is the hypothesis?

METHOD
•       How was sample size determined? (Sampling technique!),
•       What about the inclusion and exclusion criteria?
•       The procedure is rigorous and well described and the statistical analysis is correct.

-Line 93: under-16 referred twice, please check.

DISCUSSION

The discussion needs to reflect what you found, how it relates to the literature and each paragraph should be logical in sequence as at present it is a bit hard to follow. 

'As practical implications, this study showed that the older, the better ' what does this means? by what age? how do you support this result?

CONCLUSION
•       Finally, improve the conclusion section: it is important to suggest possible future studies.

Reviewer 4 Report

This study is devoted to the important issue of variation in anthropometry, body composition, and physical fitness according to age and playing positions in soccer players (from under-16 to under-19). Interestingly lean body mass was found to influence performance more than fat mass. Players' anthropometric and performance characteristics were also found to be different according to playing position.

In my opinion, this study adds some interesting observations in the area of sports sciences, promoting targeted training. The manuscript is generally well written, and the statistics seem correct (even if some verification is needed). Having said that, I would now like to turn to what I perceive the major problem of the paper to be: the equations used (Reilly et al, 2009, for fat mass; Lee et al, 2000, for muscle mass) were developed for adult individuals and are not applicable in this sample. Among the most widely used equations applicable, I suggest those of Slaughter et al (1988), but you can find other proposals in the literature. Look, for example, at what was reported by Rodriguez et al (European Journal of Clinical Nutrition (2005) 59, 1158–1166).

More minor concerns:

  • Introduction

-lines 49-51, page 2: Specify whether or not the trend described refers to the adult.

-lines 62-63, page 2: The introduction could be improved by adding more evidence from the literature about the association between training and body composition in the soccer child. See, for example, the following two studies on this topic: 

.Ateş B. Enhanced Body Composition and Physical Fitness in Prepubescent Soccer Players. J. Pedagog. Res. 2018;3:10.

.Rinaldo N. et al. Soccer training programme improved the body composition of pre-adolescent boys and increased their satisfaction with their body image. Acta Paediatr. 2016; 105:e492-5.

  • Materials and Methods

-lines 87-93, page 2: Report the mean and SD of all characters to one decimal place (as you did in the tables).

-line 88, page 2: Height should be indicated in cm, not m.

-Lines 90-92, page 2: The phrase does not sound right in English. To be reviewed.

-Figure 1: Again, the unit of measurement of stature must be the centimeter.

-Lines 134-135, page 3: the sentence "…the starting line with his." is incomplete or incorrect.

-Lines 162-163, page 4: Strangely, the skinfold thicknesses were found to have a normal distribution since these characters are known in anthropometry as the only ones that need to be normalized through a logarithmic transformation. Please, verify.

  • Results

-Figures 2, 3: an explanation of the acronyms in the histograms is necessary.

-Tables 1, 2: Change the unit of measurement of stature.

-Lines 241-246, page 7: Avoid repeating data already reported in the table.

-Table 3: Check the value of r = 0.463** because it is very different from the one reported in the text.

  • Discussion

-Line 258, page 8: Change “as they get old” to “with increasing age”.

  • Conclusions

-Lines 334-335, page 9: Add “in soccer players” after "were observed".

-Line 335, page 9: Change “the body fat” to “the body fat percentage”.

Round 2

Reviewer 1 Report

The discussion is far clearer than it was. The work is not of high quality but it is technically good. 

Reviewer 4 Report

The authors resolved all of the minor concerns I had reported. However, the most serious problem (“major”), that is the application on adolescents of equations for defining fat and muscle mass developed for the adult, was not addressed at all. On the other hand, the simplistic answer given in this regard by the authors is not acceptable “ATHLETES WITH MORE FAT MASS USING ONE EQUATION WILL HAVE MORE FAT MASS IN ANOTHER EQUATION AND THE SAME HAPPENS WITH MUSCLE MASS”. This choice is not even justifiable, as they claim, for comparison with adult athletes (which in any case does not appear in the experimental study).

Unfortunately, this is a serious methodological error that, in my opinion, prevents the work from being published in this form.  Only the application of formulas suitable to the age of the subjects and of which I had given indications in the first revision can be resolutive.

Round 3

Reviewer 4 Report

The manuscript has been revised taking into account my criticisms. I, therefore, believe that it is now publishable.